# Creating a CLOUDY-Compatible Database with CHIANTI Version 10 Data

**Chamani M. Gunasekera** *⬤, **Marios Chatzikos** *⬤ and **Gary J. Ferland** *⬤

Department of Physics & Astronomy, University of Kentucky, Lexington, KY 40506, USA
* Correspondence: cmgunasekera@uky.edu (C.M.G.); mchatzikos@uky.edu (M.C.); gary.j.ferland@uky.edu (G.J.F.)

**Abstract:** Atomic and molecular data are required to conduct the detailed calculations of microphysical processes performed by CLOUDY to predict the spectra of a theoretical model. CLOUDY now utilizes three atomic and molecular databases, one of which is CHIANTI version 7.1. CHIANTI version 10.0.1 is available, but its format has changed. CLOUDY is incompatible with the newer version. We have developed a script to convert the version 10.0.1 database into its version 7.1 format so that CLOUDY does not have to change every time there is a new CHIANTI version with an evolved format. This study outlines the steps taken by the script for this version format change. We have also found a modest number of significant changes to spectral line intensities/luminosities calculated by CLOUDY with the adoption of CHIANTI version 10.0.1. These changes are a result of improvements to collision strength data.

**Keywords:** atomic database; chianti; cloudy





## 1. Introduction

In astronomy, we cannot generally conduct direct experiments, so theoretical modeling becomes an essential tool in understanding and explaining observational results. CLOUDY is an open-source modeling software that simulates a broad range of conditions in the interstellar matter and predicts the emitted spectra using ab initio detailed calculations of microphysical processes [1].

Emission line spectra may be produced via collisional excitation followed by de-excitation of atoms and ions of various elements present in the plasmas of distant objects [2]. Hence atomic and molecular data are required by CLOUDY to conduct these detailed calculations.

CLOUDY currently incorporates three atomic and molecular databases: Stout [3], CHIANTI version 7.1.4 (referred to as Ch7 hereafter) [4], and LAMDA [5]. There have been more recent versions of CHIANTI with more accurate and extensive atomic and molecular data. However, these new versions have made major changes to the formatting of their data files, so none have been incorporated into CLOUDY thus far. CLOUDY would greatly benefit from these improvements, yet they come at the cost of modifying the source code of the software to keep up with the evolving formatting of the data. Thus, the primary goal of this paper is to reformat the latest version of CHIANTI to the format already used by CLOUDY, without having to make any changes to its source code. An ancillary objective is to keep the download size of CLOUDY manageable, requiring us to trim the database to what is essential for the operation of our code.

This paper will be arranged according to the following. In Section 1.1, we describe the CHIANTI database and its structure. Since the collisional data in CHIANTI are in Burgess and Tully space, we provide the relevant descaling equations for each of the six transition types. Our method of adapting the latest CHIANTI database to be used by CLOUDY and an analysis of the collisional data is provided in Section 2.1. Finally, in Section 3 we discuss the changes this new database has made to the test simulations in CLOUDY.

*1.1. The CHIANTI Database*

CHIANTI was originally released in 1996 [6]. It was created using observational data taken from the best available publications at the time, and theoretical estimates of data unavailable from observations. As more accurate observations and improved atomic models have become available over the years, subsequently ten CHIANTI versions have been released. Of these, we want to include the 2021 release, CHIANTI version 10.0.1 (referred to as Ch10 hereafter) [7] in the next CLOUDY release.

Ch10 has become an extensive atomic database containing energy level data, wavelength and radiative data, and electron excitation data for a large number of transitions per ion. The database is organized into three main data files for each ion. Energy level data are stored in files with extension names '.elvlc', containing both the observational and theoretical energies. The observational energies are obtained mainly from the NIST database [8]. Transition wavelengths, Einstein A values, and oscillator strengths are stored in files with the extension '.wgfa' and are obtained from the literature. For the transitions where these data are unavailable in the literature, they calculated Einstein A, and gf values using theoretical energies obtained from the literature. Lastly, excitation data containing effective collision strengths are found in the '.scups' files. These data have been gathered from the literature and are recorded in Burgess and Tully space in all versions of CHIANTI (detailed discussion and how to descale provided in Appendix A). Other auxiliary data files are available in the CHIANTI databases. However, since they are not required for any of the CLOUDY calculations, only the three file types introduced above are adapted to be used by CLOUDY and discussed in the present paper.

The Ch7 database released in 2011 is structured similarly to Ch10, with the exception of the '.scups' files. In Ch7, the electron excitation data are stored in files with extension names '.splups'. Their file format change from .splups to .scups was to better capture the structures present in the collision strength-temperature profiles for low temperatures (further details in Section 2.1). Moreover, Ch10 contains many transition levels and temperature data in the '.scups' that were not included in Ch7, making the former ∼26 times the size of the latter even without the auxiliary data files.

## 2. Ingesting a Fluid Atomic Database

*2.1. A Database Strategy*

As more detailed experimental and theoretical works are published, atomic data change. Since improving atomic data will also impact the calculations made by CLOUDY, we must have a strategy for keeping up with these evolving data sets. For our Stout data, we have scripts that easily import the ADF04[1] format. Our goal in this work is to adopt a similar strategy for the CHIANTI database. Since CHIANTI formatting undergoes significant changes from version to version, it would take some effort to modify CLOUDY to keep up with these changes. Moreover, changing the CLOUDY source code would require someone proficient in C++ atomic data objects within CLOUDY, and finding such a person is a challenging task. Instead, we developed a strategy of converting the latest version of CHIANTI to the Ch7 format (which we have used for some time [1]) using a Python script. As the CHIANTI format changes, we can easily update our script to maintain the Ch7 formatting.

CLOUDY reads in the CHIANTI data character by character of each row of data in each file, as Ch7 has columns of data that run into each other. So our reformatted database must follow the Ch7 character spacing exactly. Table A1 shows there is little change between the Ch7 and Ch10 formats for the '.elvlc' and '.wgfa' files. Reformatting these files is a simple re-organization of columns. The collisional data files require a bit more work. Unlike Ch7, which implicitly has a regularly spaced temperature grid, the grid in Ch10 is optimized to best map the data with as few points as possible. In the next section, we lay out the steps to convert the three-line Ch10 collision strengths with irregularly spaced scaled temperature into a single line with a regularly spaced grid.

We developed a Python 3 code called `chianti2oldChianti.py`. It is available at https://gitlab.nublado.org/cloudy/arrack (accessed on 13 July 2022)[2]. This repository

also contains a script to descale the BT collision strengths and temperatures in physics space for all six CHIANTI transition types (Appendix A).

*2.2. Interpolating Effective Collision Strengths*

To revert the Ch10 data in the '.scups' files, we must interpolate onto the Ch7 regularly spaced grid while still preserving the collision strength-temperature relation as closely as possible. This can be achieved by increasing the number of spline points used.

Our script does the following, recursively, for each transition:

1. First we use `scipy.interpolate.interp1d` to find a best-fit function for the log of $Y_{BT}$-$T_{BT}$ relation for each transition of each ion. We omit points with $Y_{BT} = 0$ and add them back in later.
2. The best-fit function is then used to interpolate the set of $Y_{BT}$ that corresponds to a set of evenly spaced $T_{BT}$ points. As most of the Ch7 files contained 11 spline points, we begin by using a set of 11 $Y_{BT}$ points.
3. We find another function to fit the linearly spaced data with the same method as before.
4. Then using the original set of temperature points and the new best-fit function we obtain a recalculation of the original $Y_{BT}$-$T_{BT}$ relation.
5. The error ($\chi$) is computed to reveal how well the interpolated data has preserved the $Y_{BT}$-$T_{BT}$ relation for that transition,

$$\chi = \frac{1}{N}\left(\sum_i^N \left(\frac{o_i - e_i}{e_i}\right)^2\right)^{1/2} \tag{1}$$

where,

$o_i$     *i*th recalculated $Y_{BT}$ in transition;
$e_i$     *i*th original $Y_{BT}$ in transition;
$N$      number of points in the transition in Ch10.

Since $T_{BT} = 1$ in BT space represents the $T \to \infty$ limit, $Y_{BT}(T_{BT} = 1)$ in [7] is taken to be the collision strength at the high-temperature limit. We found that this value does not always smoothly follow from the $Y_{BT}$-$T_{BT}$ profile, which then skews our fits. Fitting only the values for which $T_{BT} < 0.8$ provides much improved fits from using the value at the high-temperature limit.

6. Then we repeat the previous steps for the linear $Y_{BT}$-$T_{BT}$ relation, and use the fit that corresponds to the smaller of the two absolute relative deviations.
7. Next, a spline point is added after each iteration of this procedure that meets all of the following criteria:

    1. $\chi > 0.005$;
    2. number of spline points $\leq 60$;
    3. $\Delta\chi/\chi > 0.001$.

    The scale (linear or log) that produced the smaller error in the above step is continued to be used in the following iterations of this procedure.

We set $\chi = 0.005$, 60 splines, and 0.001 error convergence as the limits of our reformatted database. We arrived at these values with a parameter-space exploration. Values corresponding to more relaxed criteria did not yield satisfactory fits to the BT collision strength data. On the other hand, increasing the maximum allowed spline points above 60, and the $\chi$ threshold below 0.005 did little to improve our fits while increasing the size of the database larger than the original Ch10 database. As the Ch10 database is already many times larger than Ch7, we do not want the reformatting process to make it even larger. Furthermore, there are transitions for which adding more points beyond a certain number does little to improve the quality of the fit. Hence we introduce a relative $\chi$ convergence threshold of 0.001 to stop the addition of any more spline points when there is little to no improvement to $\chi$ of the interpolation. Transitions that have sharp peaks/troughs in the $Y_{BT}$-$T$ profile that are difficult to capture with equidistant step sizes benefit from this

criteria. For example, the Ni XVII 15-125 (the transitions are identified in the data files by the lower- and upper-level indices specified in the level energy file) shown in Figure 1, is one such transition in which increasing beyond 15 spline points did little to improve the fit for its peak.

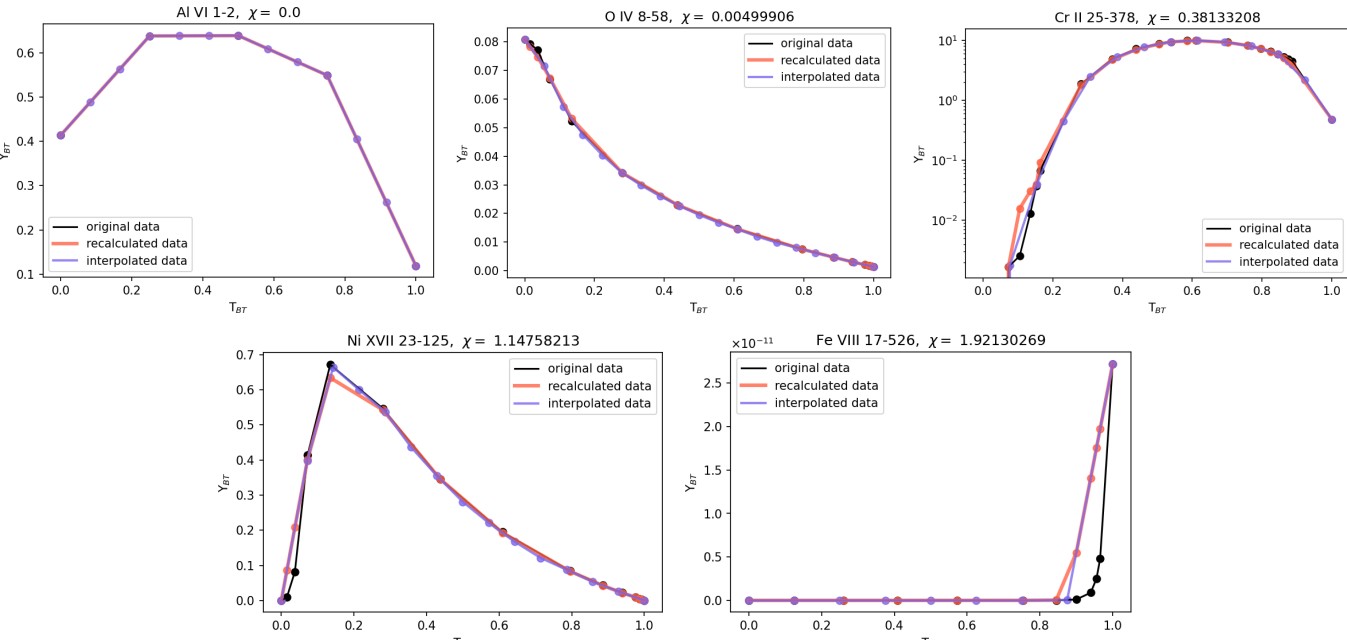

**Figure 1.** Each panel presents the collisional data of an example transition. The black plot line indicates the original data from the Ch10 database. The purple plot line indicates the evenly spaced collisional data interpolated from the original Ch10 transition. The pink plot line shows our best-fit results of the original Ch10 collision strengths.

Statistical measures for the quality of the interpolated collision strengths in our final database are presented in Figure 2. The data presented are for a truncated version of the database, further discussed in Section 2.3. The middle panel in Figure 2 reveals that only a handful of transitions have $\chi > 0.5$, and fewer still have $\chi > 1.00$. We also see that majority of the data have less than 30 splines.

Our methodology produces several transitions with fewer spline points than the minimum set limit (of 11 splines). There are two main reasons for this occurrence. First, for the transitions with five spline points, the collision strength data in Ch10 is already in an equally spaced temperature grid. Second, in the cases where we remove $Y_{BT} = 0$, the offset between the point we removed and the next $Y_{BT} \neq 0$ point is used as the minimum step size. In some cases, this minimum step size is large enough to lead to fewer than 11 spline points.

Figure 1 shows four examples of transitions with varying values of $\chi$, comparing the original set, the interpolated set and the recalculated set of $Y_{BT}$ and $T_{BT}$. Fe VIII 17-526 has the greatest $\chi$ error in the database. This is an example of a transition where the minimum step size allows for a maximum of only nine spline points. In fact, several other Fe VIII transitions with upper level 526 have this same issue resulting in $\chi > 1.0$. Ni XVII 15-125 also has a high error even though our interpolation has ended with 15 spline points. Likewise, there are several transitions in Ni XVII with upper levels of 123, 124, and 125 with 15 interpolated splines and errors >0.5. Due to the particular shape of the peak in these $Y_{BT}$-$T_{BT}$ profiles, the absolute relative convergence in $\chi$ drops below 0.001 at 15 spline points.

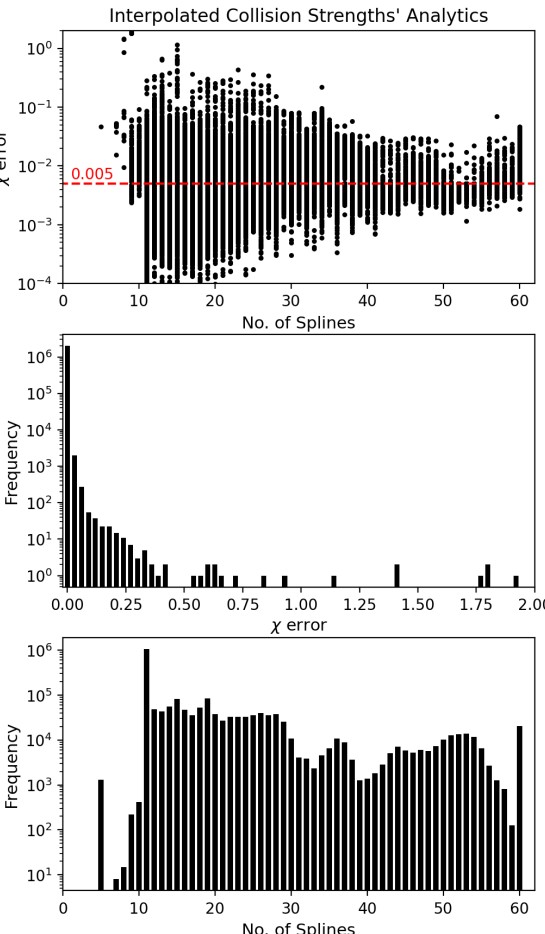

**Figure 2.** Top panel: distribution between the interpolated collision strength $\chi$ error and the number of spline points used in the interpolation, per transition. The red line in the top panel indicates the fit error limit that is set in our script that converts Ch10 format to Ch7. Middle panel: histogram of interpolated collision strength fit errors. Bottom panel: histogram of the number of spline points used in the interpolation.

### 2.3. Data Truncation

The full reprocessed database is >15 times the size of Ch7. The larger size of Ch10 is a result of most atomic models having many levels (100–1000). A large number of these levels lie above the ionization limit of that species, and CLOUDY at the moment does not process them.

Omitting these auto-ionizing levels reduces the size of the final database. We incorporated the option to make this cut into `chianti2oldChianti.py`. It follows the procedures for reformatting Ch10 as described in the above sections, but only includes levels up to the ionization limit of that ion.

Using this procedure, a truncated and reformatted version of Ch10 (hereafter referred to as NOAI) was formed, which takes up 458 MB of disk space. NOAI is $\sim 3.3\times$ smaller than the full version, $\sim 7\times$ smaller than Ch10, and $\sim 4\times$ larger than Ch07. This is a significant improvement in size and is sufficient for our purposes.

Figure 3 shows the quality of fits for the truncated database for transitions with $Y > 10^{-2}$. The recalculated and original collision strengths are presented here in physical space, converted from BT space using the equations in Appendix A. This figure reveals our method reproduces the original collision strengths decently well. The middle panel of this figure shows that all the collision strengths >100 deviate by less than 10% from the original value. Although not shown in this figure, a majority of the collision strengths in NOAI deviate by less than 1%. We also find that the collision strengths with relative

deviations $> 2\times$ and $Y > 10^{-2}$ come almost all from two ions—Ni XVII and Cr II. The ionization fraction of Ni XVII peaks at $3.98 \times 10^{6}$ K, whereas the temperature of the high deviation points in Ni XVII lies between 57,793 K and 144,035 K. Similarly, the ionization fraction of Cr II peaks at 25,119 K, while the collision strengths with high deviation for this ion lies at 1500 K. Since for both ions the ionization fraction at the temperatures of the high deviation points is far below their peaks, neither ion has an important impact on the spectral predictions. Moreover, it is only 29 collision strengths out of the total $8.6 \times 10^{6}$ points with $Y > 10^{-2}$ in the NOAI database that have these large deviations. Hence, this interpolation is sufficient for our simulations.

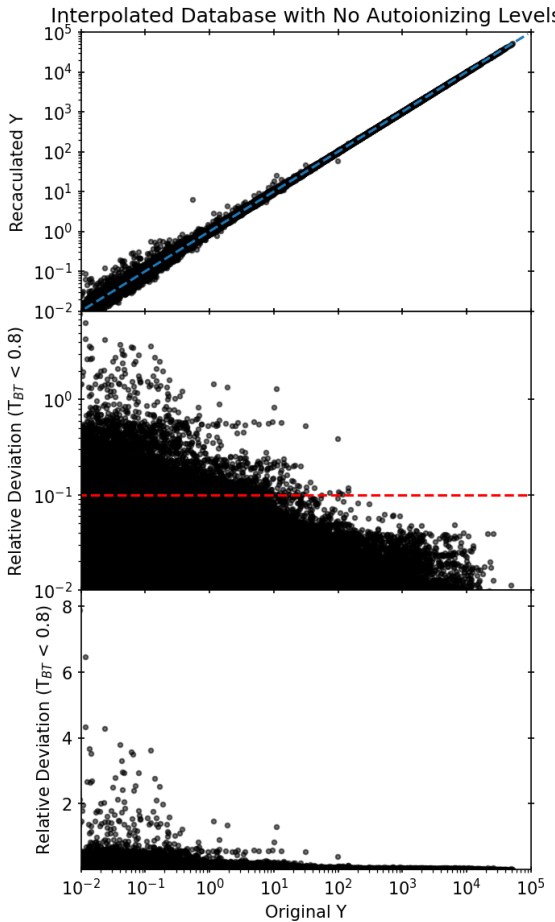

**Figure 3.** Cross-correlation between the recalculated collision strengths $>10^{-2}$ and the original Ch10 collision strengths $>10^{-2}$, both converted to physical space. The blue dashed line in the top panel is the $y = x$ plot, in our case it represents the recalculated Y and original Y that are in perfect agreement. The bottom two panels show relative deviation, which is the difference between each recalculated Y and original Ch10 Y, and divided by the latter. The red dashed line in the middle panel indicates a relative deviation of 10%. The data contained in the middle and bottom plots are the same, only differing in the scaling of the y-axis (log vs. linear).

## 3. Testing the Reformatted Database: Effect on Cloudy Models

The primary goal of this paper is to adapt the newest version of the CHIANTI atomic database to be used in the CLOUDY spectral predictions. We have accomplished this with the above-described procedures and formed a reprocessed database that can replace the current Ch7 database being used. It is now important to assess the changes this new database will induce in CLOUDY's output. A test suite built into CLOUDY monitors various observable and physical quantities in a number of different astrophysical scenarios and reports the changes that can result from alterations to the algorithm or the atomic data.

Running the CLOUDY executable test command revealed the wavelengths of 11 electron-excitation lines had changed between Ch7 and Ch10. Since CLOUDY utilizes these wavelengths in its source code, we updated these values. A summary of the wavelength changes is provided in Table 1.

**Table 1.** Wavelengths of electron-excitation transitions in the Ch10 database compared with the values in the Ch7 database.

| Ion | Ch10 Wavelength | Ch7 Wavelength |
|---|---|---|
| Al XII | 550.031 | 550.032 |
| Al XII | 568.120 | 568.122 |
| Ne VIII | 770.428 | 770.410 |
| Ne VIII | 780.385 | 780.325 |
| Ne VII | 887.293 | 887.279 |
| Ne VII | 895.191 | 895.174 |
| O VI | 1037.610 | 1037.620 |
| O IV | 1397.230 | 1397.200 |
| O IV | 1399.780 | 1399.770 |
| O IV | 1401.160 | 1404.780 |
| O IV | 1404.810 | 1404.780 |
| C IV | 1550.770 | 1550.780 |

Running the test suite with the NOAI revealed multiple changes to physical quantities as a result of changing the atomic data. In CLOUDY, if such a difference exceeds a specified tolerance it is referred to as a 'botch'. A summary of these variations is provided in Tables 2 and 3. The following is a discussion of the changes that produced the variation in the physical predictions.

**Table 2.** A list of the spectral lines that differ in intensity (the normalization line intensities used are H$\alpha$ & H$\beta$) due to collisional data changes between Ch7 and Ch10, as calculated with the time-steady simulations in the CLOUDY test suite.

| Ion | Wavelength | Transition | Time-Steady Simulations | Relative Intensity Change | Source of Change |
|---|---|---|---|---|---|
| O IV | 25.8863 μ | 1-2 | limit_lowd0.out | 0.472 | `o_4.splups` in CHIANTI 8 |
|  |  |  | limit_lowdm6.out | 0.472 |  |
|  |  |  | nlr_paris.out | 0.261 |  |
|  |  |  | pn_ots.out | 0.183 |  |
|  |  |  | pn_paris.out | 0.181 |  |
| Ne V | 24.3109 μ | 1-2 | limit_lowd0.out | −0.441 | `ne_5.splups` in CHIANTI 10 |
|  |  |  | limit_lowdm6.out | −0.440 |  |
|  |  |  | nlr_paris.out | −0.291 |  |
|  |  |  | pn_ots.out | −0.228 |  |
|  |  |  | pn_paris.out | −0.229 |  |
|  |  |  | pn_paris_cpre.out | −0.224 |  |
| Ne V | 14.3178 μ | 2-3 | limit_lowd0.out | −0.523 | `ne_5.splups` in CHIANTI 10 |
|  |  |  | limit_lowdm6.out | −0.522 |  |
|  |  |  | nlr_paris.out | −0.411 |  |
| Mg IV | 4.48711 μ | 1-2 | pn_fluc.out | −0.167 | `mg_4.splups` in CHIANTI 8 |
|  |  |  | pn_ots.out | −0.194 |  |
|  |  |  | pn_paris.out | −0.167 |  |
|  |  |  | nlr_paris_cpre.out | −0.192 |  |
|  |  |  | nlr_paris_fast.out | −0.191 |  |

**Table 3.** A list of the spectral transitions that differ in log luminosity due to collisional data changes between Ch7 and Ch10, as calculated by time-dependent test simulations in CLOUDY.

| Ion | Wavelength | Transition | Time-Dependent Simulations | Relative Log Luminosity Change | Source of Change |
|---|---|---|---|---|---|
| Fe12 | 2405.68 A | 1-2 | time_cool_cd.out<br>time_cool_cd_eq.out | 0.367<br>0.367 | `fe_12.splups`<br>in CHIANTI 10 |
| Fe12 | 2169.08 A | 1-3 | time_cool_cd.out<br>time_cool_cd_eq.out | 0.214<br>0.214 | `fe_12.splups`<br>in CHIANTI 10 |
| Fe12 | 1349.40 A | 1-4 | time_cool_cd.out<br>time_cool_cd_eq.out | 0.431<br>0.432 | `fe_12.splups`<br>in CHIANTI 10 |
| Fe12 | 1242.01 A | 1-5 | time_cool_cd.out<br>time_cool_cd_eq.out | 0.427<br>0.428 | `fe_12.splups`<br>in CHIANTI 10 |
| Fe13 | 1.07462 μ | 1-2 | time_cool_cd.out<br>time_cool_cd_eq.out | −0.223<br>−0.223 | `fe_13.splups`<br>in CHIANTI 9 |
| Fe13 | 1.07978 μ | 2-3 | time_cool_cd.out<br>time_cool_cd_eq.out | −0.315<br>−0.316 | `fe_13.splups`<br>in CHIANTI 8 |
| Fe14 | 5303.00 A | 1-2 | time_cool_cd.out<br>time_cool_cd_eq.out | −0.281<br>−0.281 | |

*3.1. Time-Steady Model Simulations*

The results of multiple simulations revealed changes to the line intensities of [O IV] $\lambda 25.8863$ μm, [Ne V] $\lambda 24.3109$ μm, [Ne V] $\lambda 14.3178$ μm, and [Mg IV] $\lambda 4.48711$ μm. The simulations with the prefix 'pn' model a planetary nebula, and those with the prefix 'nlr' model the narrow line region of an AGN. The planetary nebula model is ionized by a very hot central object, resulting in a large He II abundance. This model, a benchmark for the Paris meeting on photoionization models [9], is important for assessing the photoionization calculations performed by CLOUDY.

The variations involving electron transitions in O IV, Ne V, and Mg IV are all a result of the updated collision strength data affecting the line intensities:

- O IV: According to the review of CHIANTI 8 in [10], the collisional data from Liang et al. (2012) replaced those of Aggarwal and Keenan (2008).
- Ne V: According to the review of the Ch10 database in [7], a new model used to obtain 304 bound levels replaced a model using R-matrix calculations with only 49 levels.
- Mg IV: According to the review of CHIANTI 8 in [10], the previous CHIANTI versions contained limited data for this ion due to a lack of accuracy in the data.

As seen in Figures 4 and 5 changes to O IV, Ne V, and Mg IV collisional data affect the emissivity in these lines. Emissivity ($j_\nu$) is a function of the density of ionized atoms ($n_i(X^{(r)})$ in state i, the kinetic temperature of the gas, and the transition probabilities ($A_{ij}$),

$$j_\nu = \frac{h\nu_{ij} n_i(X^{(r)}) A_{ij}}{4\pi},\qquad(2)$$

where $\nu_{ij}$ is the frequency at the line center [2]. Since there is little to no change in the transition probabilities of these transitions between the two databases, the culprit is the population of the upper levels. The excitation rate coefficient is directly proportional to $Y_{ij}(T)$ and inversely to the square root of the temperature [11],

$$q(j \to i) = 2\pi^{1/2} a_0 \hbar m_e^{-1} \left(\frac{I_\infty}{kT}\right)^{1/2} \frac{Y_{ij}}{\omega_j}\qquad(3)$$

where $T$ is the plasma temperature, $\omega_j$ is the statistical weight of level j, $a_0$ is the Bohr radius, $m_e$ is the electron mass, and $I_\infty$ is the Rydberg constant in eV. It is also directly proportional to the de-excitation rate coefficient since one is derived from the other,

$$q(i \rightarrow j) = \left(\frac{\omega_j}{\omega_i}\right) \exp\left(-\frac{E_{ij}}{kT}\right) q(j \rightarrow i), \tag{4}$$

where $E_{ij}$ is the transition energy. Naturally, changes to the collision strengths affect both the rate of excitations and de-excitations. The bottom-most panels of Figure 4 show that the difference in temperature profiles between the two databases is very minimal and only at shallow depths of the cloud. Hence the population of the upper level is affected by variations in $Y_{ij}$, and also by changes to the collision strengths of other transitions in that ion to varying degrees.

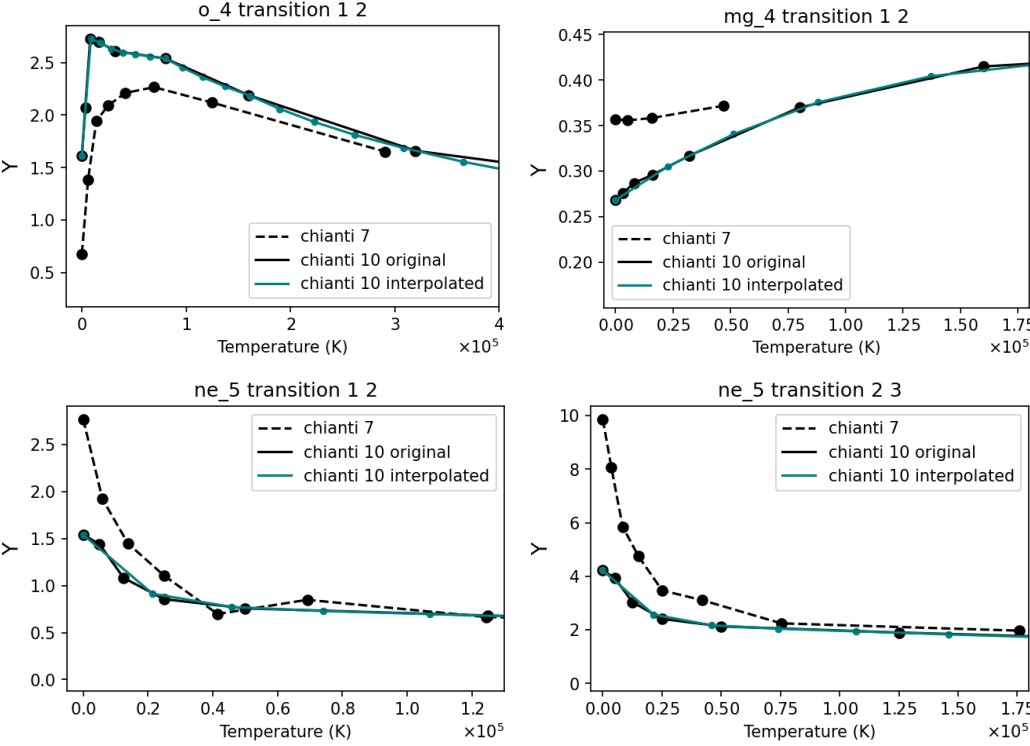

**Figure 4.** Collision strength–temperature profiles in BT space for the botched transitions in the `pn_paris` and `nlr_paris` test simulations.

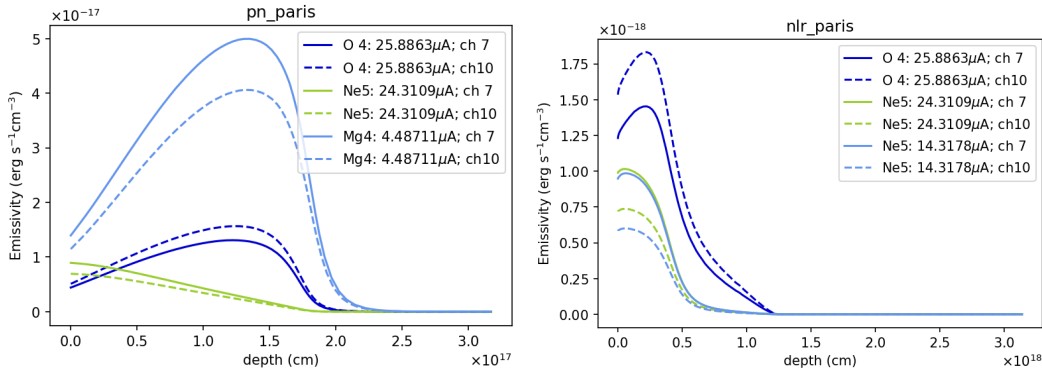

**Figure 5.** *Cont.*

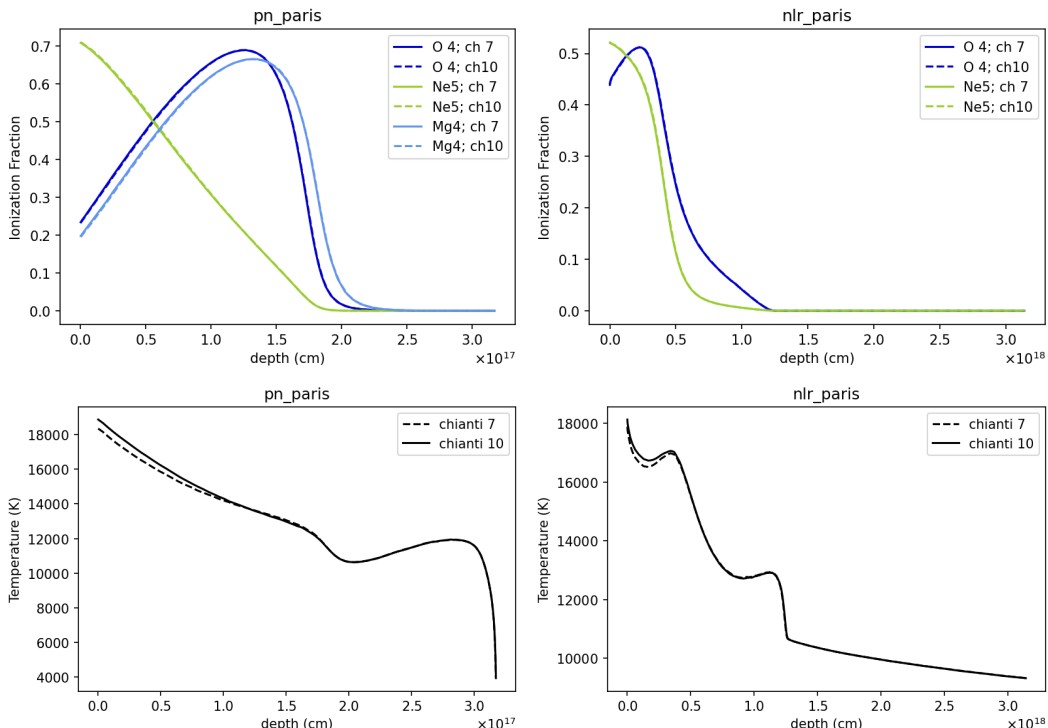

**Figure 5.** CLOUDY outputs of the `pn_paris` and `nlr_paris` test simulations.

### 3.2. Time-Dependent Model Simulations

Two time-dependent simulations reveal changes to the luminosities of spectral lines of Fe XII, Fe XIII, and Fe IV. These model time-dependent cooling of a cloud with constant density and are set to predict the time-integrated cumulative energy calculated using the mass. The simulation with the suffix 'eq' models only equilibrium cooling [12].

These luminosity variations involve transitions in Fe XII, Fe XIII, and Fe IV are all a result of a cumulative effect on the cooling mechanisms of the plasma (these variations are pertinent to studies involving effects of altering the cooling mechanisms in photoionization models, such as that presented in [13]). This is a result of the changes to the collisional data in the Fe XII transitions (collision strength changes shown in Figure 6):

- Fe XII: According to the review of the CHIANTI 8 database in [10], collisional data are obtained from the UK APAP network which includes large R-matrix calculations of 912 levels, replacing the previous R-matrix calculations of only 143 levels.
- Fe XIII: According to the review of the CHIANTI 8 database in [10], similar to Fe XII, atomic data from a larger R-matrix calculation (749 levels) replaced a smaller one (114 levels).
- Fe XIV: Collisional data for this ion have not changed since Ch7.

The temperature of the plasma and the cooling efficiency in the test simulation that contained the variations to the log luminosities are presented in Figure 7. CLOUDY outputs the total cooling as a function of temperature ($L(T)$), which has only a factor of difference from the cooling efficiency ($\Lambda(T)$),

$$\Lambda(T) = \frac{L(T)}{n_e n_p}, \tag{5}$$

where $n_e$ and $n_p$ are the electron and proton densities, respectively. We find that at $\sim 10^7$ K the cooling efficiency diverges between the simulations using the Ch7 and the NOAI databases. This results in a divergence in the temperature calculated using these two databases. We also see in Figure 7 that at temperatures following the divergence, Fe XII,

Fe XIII, and Fe XIV become the dominant coolants [14]. Furthermore, for a gas cooling freely at a rate of $\dot{M}$, the total luminosity in the line is

$$L_\nu = \dot{M}\Gamma(T_{\max}),\tag{6}$$

where,

$$\Gamma(T, T_{\max}) = \frac{3}{2}\frac{k_B}{\mu m_p}\int_T^{T_{\max}}\frac{\Lambda_\nu(T')}{\Lambda(T')}dT'.\tag{7}$$

is the total emission per unit mass in the line (e.g., [12]). The other symbols in Equation (7) are,

$\dot{M}$    mass deposition rate;
$k_B$    Boltzmann constant;
μ    mean molecular weight;
$m_p$    proton mass;
$\Lambda_\nu(T)$    frequency-integrated line cooling.

Since the luminosity of the line is a function of the cooling efficiency and temperature, the luminosity of the lines in the above electron transitions has changed as a result of changes to the collision strength data.

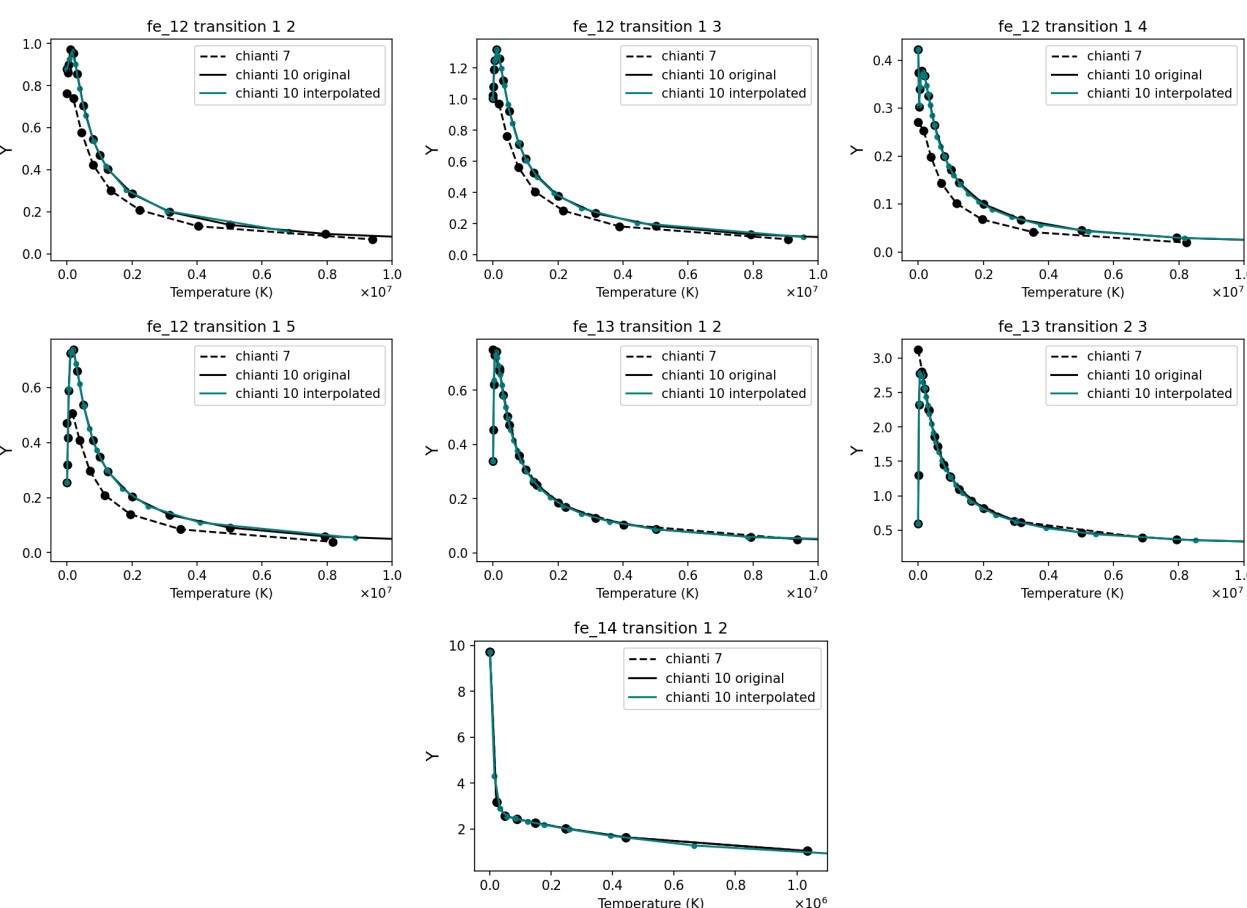

**Figure 6.** Collision strength against temperature in BT space for botched Fe XII, Fe XIII, and Fe XIV electron transitions in the `time_cool_cd` test simulation. Green plot lines indicate equally spaced temperature grid data interpolated from the original source of the Ch10 data. The original data show all but the last data point, which corresponds to the point at the high-temperature limit.

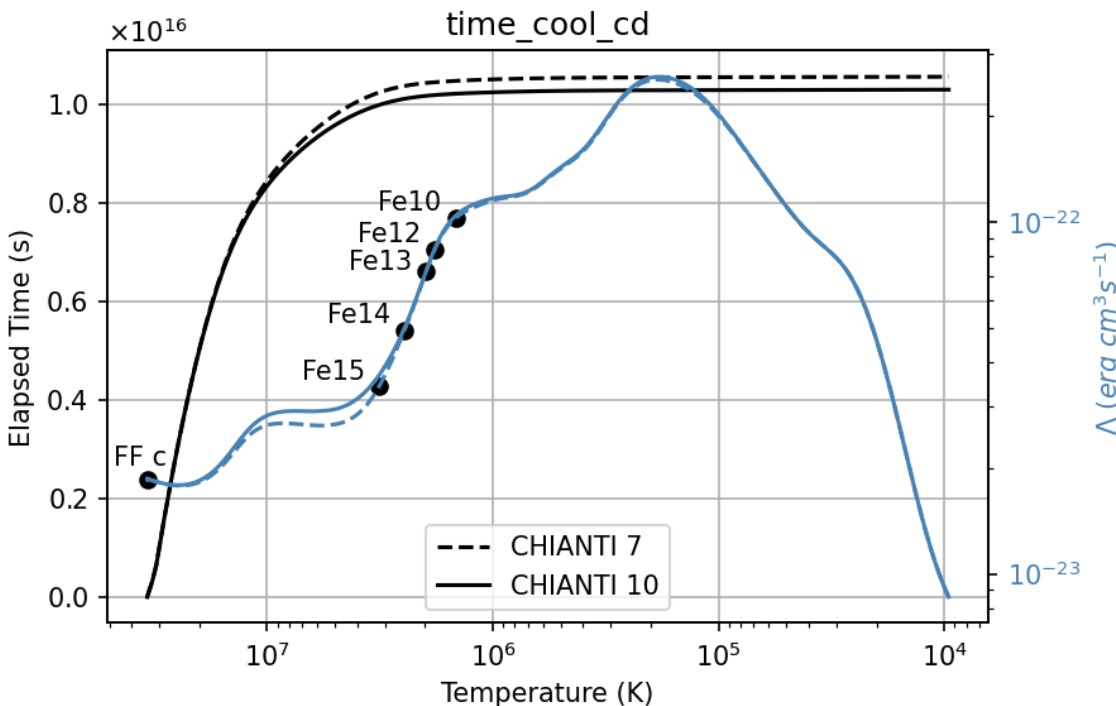

**Figure 7.** Temperature as a function of time (in black), and total cooling as a function of temperature (in blue). Dashed lines indicate the output of the simulation utilizing atomic data from the Ch7 database, and solid lines indicate the output of the simulation utilizing the reprocessed Ch10 database. The black circles indicate the dominant coolant at that temperature and the following temperatures up to the next black circle. Free-free cooling is denominated by 'FFc'.

## 4. Summary

In adopting the newest version of CHIANTI, we developed a script that will reprocess the Ch10 data to the format of the version currently used by CLOUDY-Ch7. This allows us to use Ch10 data in the CLOUDY calculations, without changing CLOUDY. As future versions of CHIANTI become available, we will be able to account for format changes when adopting to CLOUDY with only minor modifications to our external script.

The `.elvlc` and `.wgfa` files in Ch10 require only minor changes to the spacing between the data and a re-organization of the columns to be converted into the Ch7 format. In contrast, the `.scups` to `.splups` file format conversion required an interpolation of Ch10 collision strength data due to its irregularly spaced temperature grids. Finally, to reduce the size of our atomic database, we omitted all autoionizing levels. This was done since the Ch10 database includes many levels above the ionization limit which are currently not used by CLOUDY. Statistical analyses showed that the resulting truncated and reprocessed database is sufficiently small and accurate enough to be used in the microphysical calculations in CLOUDY.

Changing the collision strength data from Ch7 to Ch10 produced variations in the intensity of 10 different spectral lines predicted by CLOUDY. Three of these ten line variations were observed in six different time-steady test simulations (modeling planetary nebulae, and narrow-line AGNs). The impact of the change in collision strength data (of O IV, Ne V, and Mg IV) on the upper-level populations of the transitions resulted in significant changes to the line intensities of [O IV] $\lambda 25.8863\,\mu$, [Ne V] $\lambda 24.3109\,\mu$, [Ne V] $\lambda 14.3178\,$m, and [Mg IV] $\lambda 4.48711\,\mu$A. The remaining seven line variations were observed in time-dependent test simulations of a cooling isochoric plasma. The impact of the change in collision strength data on the cooling efficiency of the plasma resulted in significant changes to the line luminosities of [Fe XII] $\lambda 2406$, [Fe XII] $\lambda 2169$, [Fe XII] $\lambda 1349$, [Fe XII] $\lambda 1242$, [Fe XIII] $\lambda 1.07462\,\mu$, [Fe XIII] $\lambda 1.07978 \mu$, and [Fe XIII] $\lambda 5303$.

**Author Contributions:** Conceptualization, M.C. and G.J.F.; Formal analysis, C.M.G.; Funding acquisition, M.C. and G.J.F.; Investigation, C.M.G.; Project administration, M.C. and G.J.F.; Supervision, M.C. and G.J.F.; Writing—original draft, C.M.G., Writing—review and editing, C.M.G., M.C. and G.J.F. All authors have read and agreed to the published version of the manuscript.

**Funding:** C.M.G. acknowledges support by NASA grant 19-ATP19-0188. M.C. acknowledges support by STScI (HST-AR14556.001-A), NSF (1910687), and NASA (19-ATP19-0188). G.J.F. acknowledges support by NSF (1816537, 1910687), NASA (ATP 17-ATP17-0141, 19-ATP19-0188), and STScI (HST-AR-15018 and HST-GO-16196.003-A).

**Institutional Review Board Statement:** Not applicable.

**Informed Consent Statement:** Not applicable.

**Data Availability Statement:** The CHIANTI version 10.0.1 database was obtained from https://www.chiantidatabase.org/ accessed on 26 August 2020. The two different reformatted forms of CHIANTI version 10.0.1, produced in this study, `chianti_v10.0_full` without any truncations, and `chianti_v10.0_noai` without autoionizing levels, are available as compressed files that can be downloaded from http://data.nublado.org/chianti/ accessed on 7 March 2022.

**Acknowledgments:** We thank Giulio Del Zanna and the entire CHIANTI team for the providing community with such an excellent atomic database. Their work made this paper, and CLOUDY itself possible. *Software:* Arrack (https://gitlab.nublado.org/arrack accessed on 13 July 2022), Python 3.8 [15], and CLOUDY [1].

**Conflicts of Interest:** The authors declare no conflict of interest. The funders had no role in the design of the study; in the collection, analyses, or interpretation of data; in the writing of the manuscript, or in the decision to publish the results.

## Abbreviations

The following abbreviations are used in this manuscript:

| BT | Burgess and Tully |
|---|---|
| Ch10 | CHIANTI database version 10.0.1 |
| Ch7 | CHIANTI database version 7.1 |
| NOAI | Reprocessed CHIANTI v10.0.1 with no autoionizing levels |

## Appendix A. Burgess and Tully Scaling

CLOUDY utilizes collisional data from various sources for its microphysical calculations. The collisional data in the CHIANTI database, however, are scaled using the [11] (BT hereafter) method. CLOUDY has to convert the CHIANTI collisional data from BT space to physical units. Below we review the equations for BT scaling, which we use in our analysis in Section 2.3.

The BT method describes a way to scale electron-impact collision strengths of positive ions in a compact form. In this procedure, both collision strengths and temperatures are mapped onto a finite range of values, based on the type of transition [11]. For temperature, this is an interval of $(0, 1)$ for all transition types. Although the original BT publication discusses only four types of transitions (optically allowed non-zero gf, allowed small gf, forbidden, and exchange), work on CHIANTI has introduced two additional transition types. The classification of the transitions and the descaling equations are as follows:

**Type 1** Optically allowed transitions with non-zero oscillator strengths.

$$Y = Y_{BT} \ln \left( \frac{kT}{E_{ij}} + \exp 1 \right)$$

**Type 2** Optically forbidden transitions induced by an electric or a magnetic multipole interaction.

$$Y = Y_{BT}$$

**Type 3** Transition induced by exchange between incident and bound electrons resulting in a change in the spin of the ion.

$$Y = \frac{Y_{BT}}{\frac{kT}{E_{ij}} + 1}$$

**Type 4** Similar to Type 1 transition: an optically allowed transition but with a very low oscillator strength.

$$Y = Y_{BT} \ln\left(\frac{kT}{E_{ij}} + C\right)$$

**Type 5** Transition involving dielectronic recombination excitation.

$$Y = \frac{Y_{BT}}{kT/E_{ij}}$$

**Type 6** Forbidden type proton transitions.

$$Y = 10^{Y_{BT}}$$

where,

$$\frac{kT}{E_{ij}} = \begin{cases} \exp\left(\frac{\ln(C)}{1-T_{BT}}\right) - C, & \text{Transition Type 1 \& 4} \\ C\left(\frac{T_{BT}}{1-T_{BT}}\right), & \text{Transition Type 2 \& 3} \\ T_{BT}, & \text{Transition Type 6.} \end{cases}$$

and the notation is as follows,

$Y$     descaled collision strength;
$Y_{BT}$   collision strength in BT space;
$C$     scaling parameter;
$E_{ij}$    transition energy of $i \to j$ in unit K.

## Appendix B. CHIANTI File Formats

**Table A1.** Format variation from Ch10 to Ch7.

**.elvlc files**

|  | **Ch10** | **Ch7** | **Character Columns in Ch7** |
|---|---|---|---|
| Column 1 | Level Index | Level Index | 1–3 |
| Column 2 | Level Configuration | Level Configuration | 5–26 |
| Column 3 | - | Level Label String | omitted |
| Column 4 | Spin Multiplicity (2S + 1) | Spin Multiplicity (2S + 1) | 27 |
| Column 5 | Orbital Angular Momentum Symbol (L) | Orbital Angular Momentum Integer (L) | 30 |
| Column 6 | Total Angular Momentum (J) | Orbital Angular Momentum Symbol (L) | 32 |
| Column 7 | Observed Energy (cm$^{-1}$) | Total Angular Momentum (J) | 35-37 |
| Column 8 | Theoretical Energy (cm$^{-1}$) | Statistical Weight (2J + 1) | 40 |
| Column 9 | - | Observed Energy (cm$^{-1}$) | 41–55 |
| Column 10 | - | Observed Energy (Ry) | 56–70 |
| Column 11 | - | Theoretical Energy (cm$^{-1}$) | 71–85 |
| Column 12 | - | Theoretical Energy (Ry) | 86–100 |

**.wgfa files**

**Table A1.** *Cont.*

| .elvlc files | | | |
|---|---|---|---|
| | **Ch10** | **Ch7** | **Character Columns in Ch7** |
| Column 1 | Lower Level Index | Lower Level Index | 1–5 |
| Column 2 | Upper Level Index | Upper Level Index | 6–10 |
| Column 3 | Wavelength (Angstroms) | Wavelength (Angstroms) | 11–25 |
| Column 4 | gf Value (weighted oscillator strength) | gf Value | 32–40 |
| Column 5 | Einstein A (radiative decay rate) (s$^{-1}$) | Einstein A (s$^{-1}$) | 47–55 |
| Column 6 | Level Configuration | Level Configuration | omitted |
| .scups and .splups files | | | |
| Row 1, Column 1 | Lower Level Index | Z (atomic number) | 1–3 |
| Row 1, Column 2 | Upper Level Index | ion (no. of missing electrons) | 4–6 |
| Row 1, Column 3 | Energy of Transition (Ry) | Lower Level Index | 7–9 |
| Row 1, Column 4 | gf Value | Upper Level Index | 10–12 |
| Row 1, Column 5 | High Temperature Limit (K) | BT92 Transition Type | 15 |
| Row 1, Column 6 | Number of Scaled Temperatures | gf Value | 17–25 |
| Row 1, Column 7 | BT Transition Type | Energy of Transition (Ry) | 27–35 |
| Row 1, Column 8 | BT Scaling Parameter | BT92 Scaling Parameter | 37–45 |
| Row 1, Column 9+ | - | Scaled Effective Collision Strengths (BT scale) | 47+ |
| Row 2 | Scaled Temperatures (BT scale) | - | - |
| Row 3 | Scaled Effective Collision Strengths (BT scale) | - | - |

## Notes

[1] ADF04 data are available online at https://open.adas.ac.uk.

[2] This repository is named after a type of distilled spirit typically found in South Asia. The version found in Sri Lanka is made of unopened flowers from coconut palm giving it the taste of Cognac and rum with floral notes.

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
