# Peer review of "Creating a CLOUDY-Compatible Database with CHIANTI Version 10 Data"

_2674-0346, doi:10.3390/astronomy1030015_

Round 1

Reviewer 1 Report

This is a well written paper that clearly describes the authors' techniques.  

However, I do not feel that the topic is of general interest.  The authors have 

developed a set of software tools for internal use by the CLOUDY team.

I would note that the website where the code is hosted is available only to 

those with gitlab accounts.

Author Response

Thank you for your diligent review.

CHIANTI is a widely used atomic database. Two other major codes XSTAR and MAPPINGS besides CLOUDY are stuck with an older version of CHIANTI for the same reasons. We provide here a solution for all such codes with the same issue, hence why we make our script open-access.

We apologize for the previous link. We have now provided a link to the code, which should be available for public access without any login necessary. We have also edited the manuscript for all grammatical errors. All the edits are shown in red font.

Reviewer 2 Report

Astrophysical modelling is an essential tool in studying observations of stellar objects and the interpretation of those observations.  Over the last number of years, the code CLOUDY has been instrumental in such studies.  The development of this code  has been spearheaded by the third author of the present paper. 

As input to its study, CLOUDY requires the provision of as much reliable atomic and molecular data as can be be made available.  Recently, this has been provided by the database CHIANTI.  This database is periodically updated, both with additional data and with higher levels of accuracy of all the data, usually through improved calculations of that data.  One of the difficulties for users of the database is that the format of its data changes from one version to the next, sometimes substantially.  Of course the current version will not be the final one.

A new version of CHIANTI has recently become available, and the format of the data has changed from earlier versions.  Rather than modify CLOUDY each time a new version of the atomic and molecular becomes available, the present authors have chosen to provide a new code which will transform the format of the CHIANTI data into the format required by the input into CLOUDY, and it is this transition which is discussed in detail in this paper.

I consider this to be an important paper, essential for making use of the revised database in the context of CLOUDY.  The paper is well written and structured, so that users of CLOUDY can understand how the new data can be employed in the most up-to-date atrophysical modelling.

I believe this paper should be published as soon as possible.

Author Response

Thank you for your diligent review.

We have now provided a link to the code, which should be available for public access without any login necessary. We have also edited the manuscript for minor grammatical errors. All the edits are shown in red font.

Reviewer 3 Report

The paper presents an overview of atomic and molecular data bases used for an analyses of astronomical phenomena. In particular, the authors consider the databases CLOUDY incorporating various CHIANTI versions. The possible solution of a problem of an evolved format in advanced CHIANTI versions is described in the manuscript. The paper gives an advertisement of the Python code developed in the work and available at https://gitlab.nublado.org/arrack2 . Examples of incorporation of data bases are given.   

The paper materials are relevant for implementation of atomic and molecular data to investigations of astronomical events and can be considered for publication in ‘Astronomy” as materials containing relevant hints.

However, before recommending for publication, I suggest minor corrections.
The authors should at least mention the Raman spectra data.

The English writings should be improved. It is surprising to meet the sentences like –“ One of the three databases CLOUDY currently utilizes is CHIANTI version ….”

Author Response

Thank you for your diligent review.

We have now provided a link to the code, which should be available for public access without any login necessary. We have also edited the manuscript for all grammatical errors. All the edits are shown in red font. 

Raman scattering is indeed interesting, and should probably be part of future development, but it does not fall within the context of this paper. Raman scattering is currently not used by CLOUDY, nor is it used in the CHIANTI database.

Reviewer 4 Report

I follow (and participate in creation of) databases in atomic physics for some 40 years. I have seen a number of trials/ libraries/ authors/ institutions in this field. Majority of these databases end their life with the retirement of authors, with a message left for decennies “These data need an urgent update”.

The other side are automatically collected data (like climate and meteo data), with thousands of lines and a note at the end: “These data have never been evaluated by humans”

Also the amount of data in atomic physics is such that following their accumulation, and in consequence, their use, exceeds the capacities of (human) researchers. In particular, theories are able to calculate thousands of spectral lines, for conditions exceeding by several orders of magnitude “room temperature”, i.e. to produce data applicable for the whole variety of astronomical/ thermonuclear/ ultra-low temperature conditions. This makes the comprehensive databases absolutely indispensable. CHIANTI is one of such bases.

The paper presented is rather unusual: it describes procedures and codes written to make different databases compatible between them. Authors wrote codes in specific programming language to retrieve databases, adoptedthem to two different versions of CHIANTI and tested in the case of iron spectral lines.

The paper does not belong strictly either to astronomy or the atomic physics, either informatics. But it is absolutely necessary to document the historical changes in databases, and keep them readable. Obviously, an argument could be risen that such a report could be published just by the database institution. But then we come back to the point just discussed, that after some time, without the written “testimony”, the data become useless, and another instutution/ authors/ standards start to develop ex-nuovo.

Therefore I valuate the paper high, even if it is rather unusual.

Resuming, this is a technical paper describing how different databases of atomic astrophysical data may be interchanged. The paper (and the work done) is important, as makes new versions of databases compatible with earlier. The description is clear and well illustrated. Overall merit: high 

Author Response

Thank you for your review.

We have now provided a link to the code, which should be available for public access without any login necessary. We have also edited the manuscript for minor grammatical errors. All the edits are shown in red font.